# Restoration of Osteogenesis by CRISPR/Cas9 Genome Editing of the Mutated *COL1A1* Gene in Osteogenesis Imperfecta

**DOI:** 10.3390/jcm10143141

**Published:** 2021-07-16

**Authors:** Hyerin Jung, Yeri Alice Rim, Narae Park, Yoojun Nam, Ji Hyeon Ju

**Affiliations:** 1CiSTEM Laboratory, Convergent Research Consortium for Immunologic Disease, College of Medicine, Seoul St. Mary’s Hospital, The Catholic University of Korea, 222 Banpo-daero, Seocho-gu, Seoul 06591, Korea; ilovehyelin@gmail.com (H.J.); llyerill0114@gmail.com (Y.A.R.); narae5322@gmail.com (N.P.); 2Catholic iPSC Research Center, College of Medicine, The Catholic University of Korea, Seoul 137-701, Korea; 3YiPSCELL, Inc., 39 Banpo-daero, Seocho-gu, Seoul 06579, Korea; givingtreemax@gmail.com; 4Divison of Rheumatology, Department of Internal Medicine, College of Medicine, Seoul St. Mary’s Hospital, The Catholic University of Korea, 222 Banpo-daero, Seocho-gu, Seoul 06281, Korea

**Keywords:** osteogenesis imperfecta, osteoblast differentiation, induced pluripotent stem cell, *COL1A1*, gene editing

## Abstract

Osteogenesis imperfecta (OI) is a genetic disease characterized by bone fragility and repeated fractures. The bone fragility associated with OI is caused by a defect in collagen formation due to mutation of *COL1A1* or *COL1A2*. Current strategies for treating OI are not curative. In this study, we generated induced pluripotent stem cells (iPSCs) from OI patient-derived blood cells harboring a mutation in the *COL1A1* gene. Osteoblast (OB) differentiated from OI-iPSCs showed abnormally decreased levels of type I collagen and osteogenic differentiation ability. Gene correction of the *COL1A1* gene using CRISPR/Cas9 recovered the decreased type I collagen expression in OBs differentiated from OI-iPSCs. The osteogenic potential of OI-iPSCs was also recovered by the gene correction. This study suggests a new possibility of treatment and in vitro disease modeling using patient-derived iPSCs and gene editing with CRISPR/Cas9.

## 1. Introduction

Osteogenesis imperfecta (OI), a genetic disease caused by defective synthesis of type I collagen [1], is associated with mutations in the *COL1A1* and *COL1A2* genes [2,3]. Type I procollagen, a major constituent of bones, consists of two molecules of pro-α1 chain and one molecule of pro-α2 chain in a triple-helical configuration [4]. Mutation in the *COL1A1* and *COL1A2* genes results in defective synthesis of the α-chains, which in turn leads to structural disintegration of the triple helix [5,6,7]. Accordingly, clinical manifestations of OI are characterized by weakened collagen, resulting bone fragility, and multiple fractures [8]. Depending on the exact genotype, mutations in these genes can also impact other collagen-rich tissues, leading to symptoms such as tooth defects, blue sclerae, and auditory impairment [9,10]. Based on the clinical severity, radiological features, and inheritance, OI is classically divided into four types: mild (type I OI), lethal (type II OI), severe progressive (type III OI), and moderately deforming (type IV OI) [11]. Type II OI is the most severe form, which most infants with this disease do not survive [11]. On the other hand, type I OI is the mildest form of the disease that results from null mutations in the *COL1A1* and *COL1A2* gene. Type I OI patients have approximately 50% of normal type I collagen in their tissue; therefore, the patient may experience bone fractures yet show fewer deformities [12]. Type I OI patients show clinical characteristics such as osteoporosis, blue sclera, mild skeletal deformities, and easy bruising.

OI is a rare and intractable genetic disease [13,14]. Various treatments have been attempted, but they cannot achieve complete recovery. The rarity of the disease, along with the fundamental gene abnormality, has prevented the development of an effective therapy. Bisphosphonates are often used for children who have moderate to severe OI [15]. Bisphosphonates increase bone mineral density and improve bone strength and eventually reduce bone fracture and pain. Surgery, including intramedullary rodding, reduces fracture risks when long bones are bowed [16,17]. Transplantation of mesenchymal stem cells (MSCs) was attempted to treat OI only to modulate the severity of the disease [18,19]. However, it is important to have better understanding of the disease to seek treatment. To date, clinical therapy or understanding of the disease is limited, and further studies are required.

One strategy for overcoming the shortage of biomaterial for research on rare diseases is the production of patient-derived induced pluripotent stem cells (iPSCs) [20,21]. Patient-derived iPSCs can recapitulate disease phenotypes in vitro; therefore, it is useful for understanding the disease and developing novel drug therapies through modeling of rare diseases.

Along with the development of iPSCs, CRISPR/Cas9 technology has been actively used to study genetic diseases in the recent years [22,23,24]. Correction of a defective gene using CRISPR/Cas9 can reveal the pivotal biological function of the specific gene. Monogenic diseases such as OI are plausible candidates for treatment using genome editing technology. In this study, we established OI patient-derived iPSC clones and corrected the mutated gene using CRISPR/Cas9. Osteogenic differentiation using OI-iPSCs mirrored the characteristics of the disease, and the correction of the collagen gene recovered the impaired differentiation ability. Here, we present new possibilities in iPSC-based in vitro study and precise gene correction on hereditary bone disease of OI.

## 2. Experimental Section

### 2.1. Ethical Approval

All procedures involving animals were performed in accordance with the Laboratory Animals Welfare Act, the Guide for the Care and Use of Laboratory Animals, and the Guidelines and Policies for Rodent Experimentation provided by the Institutional Animal Care and Use Committee of the School of Medicine of The Catholic University of Korea. The study protocol was approved by the Institutional Review Board of The Catholic University of Korea (CUMC-2016-0291-02) from the Catholic iPSCs Center.

### 2.2. PBMC Isolation from OI Patients

Blood was collected in heparin-coated tubes and diluted with phosphate-buffered saline (PBS). Diluted blood samples were then centrifuged through a Ficoll gradient (Catalogue number, 17-1440-03, GE Healthcare, Little Chalfont, Buckinghamshire, UK) for 30 min at 850× *g*. PBMCs were collected and transferred to a new tube. After wash with PBS, peripheral blood mononuclear cells (PBMCs) were resuspended in StemSpan medium (9805, STEMCELL Technologies, Vancouver, BC, Canada) supplemented with CC110 cytokine cocktail (8697, STEMCELL Technologies). Prior to use in iPSC generation, cells were maintained for 5 days at 37 °C in an atmosphere containing 5% CO_2_ [25].

### 2.3. Generation and Maintenance of OI-iPSCs

Reprogramming of iPSCs was performed as previously described [25]. PBMCs were collected as described above. Cells were transduced with Sendai viral particles generated using the CytoTune-iPS Sendai Reprogramming Kit (A16518, Life Technologies, Carlsbad, CA, USA). Cells were maintained in Essential 8 (E8) medium in vitronectin-coated dishes.

### 2.4. Construction of RGEN, ssODN and Surrogate Reporter Gene

RNA-guided endonuclease technology (RGEN) target sequences were designed using the ToolGen optimized CRISPR validation service (http://www.toolgen.com/ accessed on 29 May 2016). RGEN 3 and 4 had higher out-of-frame scores than RGEN 1 and 2. However, validation with T7E1 assay confirmed that the polymerase chain reaction (PCR) bands for RGEN 1 and 2 were clearer than those for RGEN 3 and 4 relative to the positive control. Healthy donor DNA was ordered from Macrogen. For the purpose of correcting the *COL1A1* c.2523delT mutation, the single-stranded oligodeoxynucleotides (ssODNs) was designed as a 48nt homology arm at the mutant site on both sides of the residue expected to be cleaved by the RGEN. Surrogate reporter systems were used to evaluate the efficiency of CRISPR/Cas9 in target mutations [26]. OI-iPSCs were transfected with RGEN, ssODN, and surrogate reporter plasmid constructs using the Neon 10-μL transfection kit (Life Technologies) and performed according to the manufacturer’s protocol.

### 2.5. T7 Endonuclease 1 (T7E1) Mismatch Detection Assay

T7 endonuclease 1 (T7E1) assay is a widely used method to analyze the editing efficiency of CRISPR/Cas9 and to identify cell clones containing mutations at target sites. We performed a T7E1 assay to detect in/del mutations. Genomic DNA was extracted using Exgene™ Cell SV kit (cat.no 106-101), and target regions were amplified by PCR (see Table 1 with High Fidelity (ThermoFisher, Waltham, MA, USA), according to manufacturer’s protocol. PCR products were denatured at 95 °C for 4min and re-annealed at −2 °C per second temperature ramp to 85 °C, followed by a −0.1 °C per second ramp to 25 °C. The heterocomplexed PCR product (10µL) was incubated with 10X T7E1 buffer (2 µL) and T7E1(0.3 µL) at 37 °C for 20 min. PCR products were electrophoresed on a 0.8% TBE agarose gel.

### 2.6. Osteogenic Differentiation of OI-iPSCs

In vitro osteogenic differentiation used in this manuscript was followed by the protocol that we used in our previous studies [27,28]. iPSCs were seeded onto a 0.1% gelatin-coated dish (1 × 10^5^ cells per well of a 24 well plate) at 37 °C, in 5% CO_2_, and under 95% humidity. When iPSCs were 80% confluent, usually in 2–3 days, the medium was replaced with osteogenic differentiation medium (ODM). ODM was prepared by supplementing DMEM-LG (Dulbecco’s modified Eagle medium, low glucose) with 15% FBS, 100 nM dexamethasone, 10 mM β-glycerol-2-phosphate disodium salt, and 50 µg/mL ascorbic acid. The medium was replaced every 2 days, and cells were cultured for a total of 3 weeks.

### 2.7. Alizarin Red S Staining

Alizarin Red S staining was performed to confirm calcium deposition during OB differentiation. To determine the degree of calcium deposition on day 0, 7, 14, and 21, differentiated cells were washed with sterile 1× PBS. Alizarin Red S solution was added, and samples were stained for 30 min in room temperature (RT). Thereafter, the non-specifically stained portion was rinsed several times with sterilized water. The stained samples were observed under an optical microscope. To quantify the stained Alizarin Red levels, samples were reacted with 10% (*w*/*v*) cetylpyridinium chloride solution at RT for 30 min, and the absorbance of the supernatant was measured at 595 nm using a microplate reader.

### 2.8. Von Kossa Staining

Von Kossa staining was performed to confirm the presence of calcified nodules. After washing with PBS, cells were fixed with 4% PFA for 20 min. After washing with tap water, samples were treated with 5% silver nitrate solution, exposed to UV light for 40 min, and then washed with tap water. Finally, samples were incubated with 5% sodium thiosulfate for 5 min and then washed again with tap water. The stained samples were observed under an optical microscope.

### 2.9. OsteoImage Mineralization Assay

The hydroxyapatite portion of the deposited bone-like nodule was identified using the Osteoimage Mineralization Assay (PA-1503, Lonza, Basel, Switzerland) and performed according to the manufacturer’s protocol.

### 2.10. Real-Time Polymerase Chain Reaction (PCR)

Total RNA was isolated using Trizol Reagent (15596026, Invitrogen, Carlsbad, CA, USA). The cDNA was synthesized using the RevertAid First Strand cDNA Synthesis Kit (K1621, ThermoScientific, Waltham, MA, USA) according to the manufacturer’s protocol (Invitrogen). The relative mRNA levels were normalized with the band pixel intensity of GAPDH. The used primers are shown in Table 2. This assay does not discriminate wild-type (WT) and mutant *COL1A1* mRNA.

### 2.11. Immunofluorescence Staining

Immunofluorescence was performed to confirm the expression of pluripotency markers. Cells were fixed with 4% formaldehyde for 10 min and washed twice in 0.1% PBST (1× PBS containing 0.1% Tween-20). Samples were then permeabilized for 10 min in PBS containing 0.1% Triton X-100 (Sigma-Aldrich, St. Louis, MO, USA). After blocking for 40 min with PBS containing 2% bovine serum albumin (2% PBA), cells were treated with primary antibody and incubated at RT for 2 h. After the exposure with primary antibodies, cells were washed several times with 0.1% PBST. Sections were then incubated with Alexa Fluor® 594–conjugated goat anti-rabbit IgG (H+L) antibody (A11037, Molecular Probes, Eugene, OR, USA) and Alexa Fluor® 488–conjugated goat anti-mouse IgG (H+L) antibody (A11029, Molecular Probes) for 40 min at RT; both secondaries were diluted 1:200 in PBS. Nuclear DNA was stained with 4′,6-diamidino-2-phenylindole (DAPI; 10236276001, Roche, Basel, Switzerland).

Immunofluorescence was also performed to confirm the expression of anti-collagen I in HC-iPSCs, OI-iPSCs, and geOI-iPSCs. Cells were fixed and permeabilized as described above. After blocking for 40 min with 10% normal goat serum containing 1% bovine serum albumin (BSA; Sigma-Aldrich, St. Louis, MO, USA), samples were incubated for 1 h at RT with anti-collagen I antibody (ab34710, Abcam, Cambridge, UK) diluted in 5% normal goat serum containing 1% PBA. After washing, sections were incubated for 40 min at RT with Alexa Fluor® 594–conjugated goat anti-rabbit IgG (H+L) antibody in PBS. Nuclear DNA was stained with DAPI.

### 2.12. Western Blot Analysis for Type I Collagen

Western blot analysis was performed to confirm the expression of type I collagen and osteocalcin in OB derived from HC-iPSCs, OI-iPSCs, and editOI-iPSCs. Cells were washed with PBS and lysed with NP-40 lysis buffer supplemented with protease inhibitors (10 mM leupeptin and 100 mM PMSF). Cell lysates were centrifuged at 13,000× *g* for 20 min at 4 °C, and Bradford assay was performed. Proteins were separated on 8% SDS-PAGE gels and transferred to nitrocellulose membranes. Membranes were blocked with 5% skim milk in PBST for 1 h and incubated with anti-type I collagen antibody (1:1000, C-terminal telopeptide, GTX82720, GeneTex, Irvine, CA, USA) and anti-osteocalcin antibody (1:500, sc-30044; Santa Cruz Biotechnology, Dallas, TX, USA) at 4 °C. Membranes were then washed 3 times with a 0.1% PBST and incubated for 40 min with a peroxidase conjugated affinipure goat-anti-rabbit IgG (1:5000, 111-035-144, Jackson Laboratory, Bar Harbor, ME, USA). After 4 washes with 0.1% PBST, the protein-of-interest was detected using a western blotting imaging analyzer LAS 1000.

### 2.13. Sample Preparation for TEM

TEM was performed to observe type I collagen structure. Cells were washed three times with DPBS and then fixed in 2.5% glutaraldehyde in 0.1 M cacodylate buffer (DukSan, Seoul, Korea) overnight. Cells were then washed in 0.1 M sodium cacodylate buffer, and post-fixation was performed for 1 h in a solution of 1% osmium tetroxide in 0.1 M cacodylate buffer (pH 7.4). Subsequently, samples were gradually dehydrated in ethanol and washed with propylene oxide for 15 min. Propylene oxide and epoxy resin were mixed and allowed to infiltrate the cell samples. Finally, samples were incubated overnight with eponate at 65 °C. Samples were then cut using a Leica Ultracut microtome (Leica, Deerfield, IL, USA) and mounted on amorphous-carbon-coated Cu TEM grids.

### 2.14. Teratoma Formation

Immunodeficient nude (NOD/SCID) mice were purchased from Jackson Laboratories, USA. geOI-iPSC clone (1 × 10^6^ cells) were suspended in 30 μL of medium-Matrigel (BD) mixture (DMEM/F12 medium: Matrigel 1:1). Anesthetized mice were disinfected three times with 70% ethanol and povidone. The skin and peritoneum were incised, and the testicles were held with tweezers. The cell mixtures were injected into subcutaneous and testis capsules in 7-week-old NOD mice. The peritoneum and skin were sutured with absorbable sutures. Animals were monitored for tumor growth for 10–12 weeks. After tumors were removed, samples were fixed in 4% paraformaldehyde and embedded in paraffin. Sections (4-μm thick) were subjected to H&E as follows: Sections were incubated in Harris’s hematoxylin solution for 10 min, and excess hematoxylin was removed by adding 1% HCl in 70% (*v*/*v*) alcohol for 5 s, followed by a 0.5% ammonium water wash. Slides were then stained in Eosin Y (1% (*w*/*v*)) for 90 s with a subsequent wash in running tap water for 1 min. Sections were mounted on clear glass slides and covered with thin glass coverslips.

### 2.15. Label-Free Quantification

Cells were centrifuged at 4000× *g* at 4 °C for 20 min. Cells were resuspended in 1 mL of lysis buffer (50 mM Tris, 100 mM NaCl, 20 mM DTT, protease inhibitor, and 1× phosphatases, with pH 7.5 buffer). The samples were sonicated for 5 min at 4 °C, and lysis buffer was replaced by 50 mM ammonium bicarbonate at pH 8.0 using Amicon™ Ultra-15 Centrifugal Filter Units. The protein samples were precipitated in 10% (*w*/*v*) TCA, and the pellets were solubilized in 50 mM ammonium bicarbonate and 1% (*w*/*v*) RapiGest SF (Waters). After protein extraction, quantities were determined by Bradford assay. The proteins were quantified, and their integrity was confirmed by SDS-PAGE. All LC-MS/MS analyses were performed on an LTQ Orbitrap XL mass spectrometer (Thermo Scientific, San Jose, CA, USA). A triplicate MS run was performed for a total of three samples of HC, OI and editOI, and label-free quantification (LFQ) analysis was performed with SwissProt Homo sapiens DB using the results.

### 2.16. Statistical Analysis

All experiments were repeated three or more times. The results are shown as mean and standard error of the mean. Statistical analyses were performed using GraphPad Prism Software, 5.0 (GraphPad, San Diego, CA, USA). ANOVA was used to analyze non-parametric quantitative datasets, and the one-tailed *p*-value was calculated. *, *p* < 0.01; **, *p* < 0.005; and ***, *p* < 0.001 indicated statistical significance. hiPSCs were subjected to at least three independent differentiations, and all experiments were replicated.

## 3. Results

### 3.1. Generation of Osteogenesis Imperfecta Patient Derived iPSCs

In this study, we were able to obtain blood sample from one type I OI patient. We confirmed that a T nucleotide was deleted (c.2523delT) from exon 36 of the *COL1A1* gene, replacing the 842nd amino acid (glycine) with alanine and causing a frameshift mutation affecting 266 downstream amino acids (Table 3). The mother of the patient harbors the same *COL1A1* mutation (c.2523delT) (Figure 1A). The OI patient participating in this study exhibited bone fragility, frequent fractures, and bone deformations that were observed through X-ray as well as clinical symptoms, such as blue sclera (Figure 1B). Induced pluripotent stem cells (iPSCs) were generated from peripheral blood mononuclear cells (PBMCs) from healthy controls (HC) and the OI patient (Figure 1C). Sequence analysis was performed from iPSCs derived from the OI patient-derived samples (Figure 1D). Different clones were established from the OI patient-derived iPSCs for this experiment. OI-iPSCs showed the same *COL1A1* mutation as the patient-derived primary cells confirmed by Sanger sequencing (data not shown). Both OI-iPSCs and HC-iPSCs maintained morphological characteristics unique to iPSC colonies, and their undifferentiated state was confirmed by alkaline phosphatase staining (Figure 1E). Expression of the pluripotency-associated transcription factors, such as *OCT3/4, SOX2, NANOG, LIN28*, and *TDGF1,* confirmed high pluripotency (Figure 1F). In addition, real-time RT-PCR analysis was performed on the expression of *OCT3/4*, *SOX2,* and *NANOG* in OI-iPSC and HC-iPSC (Figure 1G). The list of primers used in this study is provided in Table 2. Pluripotency of OI- and HC-iPSCs was further confirmed by flow cytometry analysis, showing expressions of OCT4, TRA-1-60, and SSEA-4 (Figure 1H). Pluripotency of OI- and HC-iPSCs was further confirmed by flow cytometry analysis, showing expressions of OCT4, TRA-1-60, and SSEA-4 (Figure 1I). As a result, it was confirmed that there was no difference between the clones of HC- and OI-iPSC. The similar expression of pluripotent markers in cell lines from each group also indicates that the defect in OI-iPSCs did not affect the pluripotency of the generated patient-derived stem cells. In conclusion, we confirmed the disease characteristics of the patient-derived sample and successfully generated OI-iPSCs.

### 3.2. OI Patient-Derived iPSCs Exhibit Delayed OB Differentiation

Osteoblast (OB) differentiation of OI-iPSCs and HC-iPSCs was performed to confirm if OI-iPSCs have lower ability to differentiate into bone. For differentiation into OBs, we induced OB differentiation with a osteogenic differentiation medium (ODM) (Figure 2A). OBs differentiated from HC-iPSCs showed a significant increase in mineralized nodule formation during OB differentiation, whereas OBs differentiated from OI-iPSCs showed no increase in mineralized nodules (Figure 2B and Appendix A). The formation of mineral nodules in OB differentiated from HC-iPSCs and OI-iPSCs was evaluated by Alizarin Red S, von Kossa, and OsteoImage staining. Alizarin Red S staining is a method commonly used to identify calcium containing osteocyte in differentiated OBs. Von Kossa staining is commonly used to observe the presence of calcium phosphate in differentiated OBs, and silver ions react with phosphate to produce black precipitates. Mineralized matrix was quantified with an OsteoImage™ mineralization assay, and the hydroxyapatite portion of the mineralized matrix was assessed qualitatively by fluorescence microscopy. Calcium deposition and matrix calcification were evaluated on days 7, 14, and 21 (Figure 2B and Appendix A). Calcified nodules appeared red by Alizarin Red S staining, brown by von Kossa staining, and fluorescent green in the OsteoImage assay. On day 21 of OB differentiation, OI-iPSCs contained fewer calcium deposits and a smaller calcification area than HC-iPSCs (Figure 2B). Staining results quantified using the ImageJ software confirmed that OI-iPSCs had less OB differentiation ability (Figure 2C and Appendix A). Quantitative RT-PCR analysis of mRNA expression levels of *RUNX2*, *COL1A1,* and *Osteocalcin (OCN)* genes was performed in OBs differentiated from HC-iPSCs and OI-iPSCs. In OBs differentiated from HC-iPSCs, the expression of *RUNX2* and *OCN* mRNA was significantly increased compared to that of OBs differentiated from OI-iPSCs (Figure 2D). We observed the collagen fibril structure from OI-iPSCs by transmission electron microscopy (TEM). The abnormal expression levels of type I collagen in OI-iPSCs was confirmed by TEM images (Figure 2E). Through these results, it was confirmed that OI-iPSCs-derived OB can reduce normal collagen levels in vitro or cause structural defects in collagen fibers. This suggests that damaged bone formation can be reproduced.

### 3.3. Correction of COL1A1 Mutation by CRISPR/Cas9

We used CRISPR/Cas9 to edit the *COL1A1* mutation in OI-iPSCs. As noted above, a deletion (c.2423delT) in exon 36 of the *COL1A1* gene caused a frameshift mutation affecting 266 downstream amino acids. We designed an exogenous ssODN template encoding homology arms spanning the target site, and RG2 was constructed to target the mutation in exon 36 (Appendix A). RNA-guided endonucleases (RGENs) uses short guide RNA (gRNA) to recognize DNA, bind endonuclease, and induce site-specific cleavage [29]. To select the single guide RNAs (sgRNAs) that can be recognized by the CRISPR/Cas9 enzyme at the *COL1A1* gene site, each exon region was inserted into CRISPR RGEN tools. Thus, four sgRNAs were selected. Four RGENs were selected, and all four RGENs had 1-0-0 mismatches, which indicates lower probabilities of off-target reactivity [30]. A list of primers is provided in Table 1. T7E1 assay was used to test the efficiency of the four RGENs. Clear bands were observed for RGEN 1 and 2 relative to the positive control. Based on this result, these two RGENs were selected for use in subsequent experiments (Appendix A). We confirmed the transfection efficiency of the red fluorescent protein (RFP) reporter gene in HEK 293T cells, which is constitutively expressed by the CMV promoter (PCMV) (Appendix A). The sgRNA target site and primers used for the T7E1 assays are shown in Table 4. To promote repair and gene conversion via homology-directed repair (HDR), a donor template was co-transfected with the CRISPR/Cas9 system (Cas9, RGEN, and surrogate vector) in iPSCs (Figure 3A and Appendix A). Also, we used a surrogate Reporter that expresses a hygromycin-resistant protein-EGFP fusion protein only when frame-shifting indels are generated in the target sequences of the reporter by nuclease activity. Over the 2–3 days after transfection, cells expressed green fluorescent proteins (GFPs) and RFPs (Figure 3B, upper panels). GFP is not expressed without RGEN activity because the GFP sequence is out of frame, and there is a stop codon before it. As a result, GFP signal can only arise from the correct, in-frame insertion at the *COL1A1* locus. Thereafter, stable, GFP-positive cells were obtained by selection with hygromycin B. The cells were passaged and maintained (Figure 3B, lower panels). Genes edited were determined by T7E1 assays of genomic DNA from unselected or hygromycin B (GFP+) selected OI-iPSCs (Figure 3C). We confirmed successful correction in the deleted target site through sequencing (Figure 3D). Gene-edited OI-iPSCs (editOI-iPSCs) retained the morphological characteristics unique to iPSC colonies; their undifferentiated state was confirmed by alkaline phosphatase expression (Figure 3E), and positive expression of pluripotency markers was confirmed in the edited cells (Figure 3F–H). Teratoma formation was observed eight weeks after subcutaneous injection of iPSCs in NOD/SCID mice, and the three germ layers were confirmed in the generated teratoma tissues (Figure 3I). Karyotype analysis of the gene-edited patient-derived iPSCs revealed the same a normal male karyotype of 46, XY as the original cells (Figure 3J). Taken all together, we successfully gene-edited the *COL1A1* mutation in OI-iPSCs and generated geOI-iPSCs using the CRISPR/Cas9 system. We also confirmed that the generated geOI-iPSCs maintained their stemness and genetic characteristics.

### 3.4. Recovery of Collagen Fibril Structure of Bone Formation in Gene-Edited OI-iPSCs

We compared OB differentiation with geOI-iPSCs, OI-iPSCs, and HC-iPSCs (Figure 4A and Appendix A). HC-iPSCs and geOI-iPSCs underwent differentiation and formed mineralized nodules starting on day seven, which was confirmed by Alizarin Red S, von Kossa, and OsteoImage staining (Appendix A). Mineralization reduced in OBs differentiated from OI-iPSCs was recovered in geOI-iPSCs, which showed similar levels to that of HC-OBs. Alizarin Red S staining showed that OBs differentiated from OI-iPSCs showing no deposition of calcification compared to OBs differentiated from HC-iPSCs and geOI-iPSCs (Figure 4A and Appendix A). The deposition of calcium phosphate by von Kossa staining showed that OBs differentiated from HC-iPSCs and geOI-iPSCs were strongly deposited compared to OBs differentiated from OI-iPSCs (Figure 4A and Appendix A). Similar to Alizarin Red S and von Kossa staining, the amount of green fluorescent staining was strongly deposited in OBs differentiated from HC-iPSCs and geOI-iPSCs (Figure 4A and Appendix A). The mineralized nodules and staining area of Alizarin Red S, von Kossa staining, and OsteoImage™ mineralization assay were quantified (Figure 4B). We measured representative Pro-Collagen 1 and Osteocalcin as biochemical markers of bone metabolism to observe the changes in biochemical bone metabolism caused by collagen mutations. Pro-Collagen 1 and Osteocalcin levels in OB supernatant were measured on day 21 of OB differentiation (Figure 4C). Measurements of Pro-Collagen 1 accounts for 90% of the total protein in the bone organic matrix and is a typical osteoblast marker [31]. The level of Pro-Collagen 1 was higher in editOI compared to OI. In addition, Pro-Collagen 1 levels in HC and editOI showed similar expression, but there was no significant difference between HC and editOI (Figure 4C, upper panel). Osteocalcin, the major protein involved in the formation of mineralized matrix, was present at significantly lower levels in the supernatant of OI-iPSCs (Figure 4C, lower panel). We reconfirmed that OCN was more abundant in editOI-iPSCs than OI-iPSCs using western blot analysis (Figure 4D). Interestingly, COL1A1 expression was increased in OI and decreased after gene editing (Figure 4D). Western blots band were quantified with Image J software and normalized. (Figure 4E). Quantitative RT-PCR analysis of COL1A1 mRNA expression levels was performed in OBs differentiated from HC, OI, and editOI-iPSCs (Figure 4F). COL1A1 mRNA levels were similar between the OBs differentiated from HC, OI, and editOI-iPSCs and had no significant difference. The expression of COL1A1 was confirmed by immunofluorescence staining (Figure 4G). As a result of immunofluorescence staining, uniform distribution of type I collagen was observed in OBs differentiated from HC-iPSCs, but abnormal distribution was observed in OBs differentiated from OI-iPSCs. To quantify abnormality distribution, intracellular type 1 collagen was evaluated (Figure 4H). Type I collagen area was quantified, confirming that OB differentiated from editOI, and HC-iPSCs contained comparable levels of collagen that were significantly greater than those in OB differentiated from OI-iPSCs (Figure 4H). Ultrastructural morphological analysis was performed using transmission electron microscopy (TEM) to analyze the structure of collagen fibers of HC-iPSCs, editOI-iPSCs, and OI-iPSCs. Normal collagen fibers with a twisted structure were observed in editOI-iPSCs. ECM with lower collagen fibril density and poorly formed fiber morphology was observed in OI-iPSCs (Figure 4I). These results confirmed that gene editing using the CRISPR/Cas9 system restored the reduced expression of type I collagen in OI-iPSCs.

### 3.5. Label-Free Quantitative Proteomic Analysis of OI-OBs and Gene-Edited OI-OBs

To characterize HC-, OI-, and editOI-OBs more comprehensively, we performed label-free quantitative (LFQ) LC-MS/MS analysis. LFQ analysis was performed using the SwissProt Homo sapiens Database, and a total of 563 proteins were quantified. A list of identified proteins can be found in supplementary information. The hierarchical clustering of proteins showed differential expression between OI-OBs and editOI-OBs. The different levels of abundance were visualized and shows that HC-OBs and editOI-OBs have the same expression pattern (Figure 5A). Among these proteins, four proteins with high Score Sequest HT in editOI-OBs (Figure 5B) and four proteins with high Score Sequest HT in OI-OBs were identified (Figure 5C). All four proteins that had high score Sequest HT in editOI-OBs showed a similar expression pattern; the expression was decreased in OI-OBs, and the expression was restored by gene editing, which resulted in similar levels to that of HC-OBs.

## 4. Discussion

OI is a genetic disorder caused by defects in the formation of type I collagen [32], characterized by fragility and multiple fractures of the bone. OI is predominantly caused by an autosomal dominant mutation of type I collagen and is known to be associated with mutations of the *COL1A1* or *COL1A2* genes involved in type I collagen synthesis [33,34,35]. This pro-collagen chain exists in a triple helix structure consisting of two pro-α1 chain molecules and one pro-α2 chain molecule. It has a unique helix structure that is twisted together by the repetition of glycine (G)-X-Y units [36]. The most important amino acid that forms the G-X-Y helical structure is glycine. When it is replaced by another amino acid, structural defects in the helical structure occur and fails to form a normal functioning procollagen chain. One abnormal chain can result in an abnormal triple helix structure even when one chain is normal. Also, it is known that the disease appears due to a problem with the formation of the type I collagen triple helix [37,38]. Many mutations of glycine have been reported in OI patients [37,39,40].

Since the clinical features of OI are relatively diverse, Sillence et al. [39]. proposed a classification system suggesting four major types of OI. This classification system is based on signs and symptoms, such as the presence or absence of clinical features, such as tooth defects, blue sclerae, or auditory impairment. Recently, a new classification system has been proposed in which the subtypes of OI are classified in more detail [41]. Most importantly, however, the quality of type I collagen is the most important characteristic between each type of OI. In this study, we investigated the effect of abnormal collagen structure on OB production and mineralization using OI patient-derived iPSCs. We produced three different iPSC clones from the PBMC of one OI patients using Sendai virus (SeV)-reprogramming vectors. The generated cell lines showed similar levels of pluripotency as HC-iPSC. As a result, OB differentiation and formation of bone matrix were suppressed in OI-iPSCs, resulting in abnormal mineralization. Our results indicate that type I collagen production in OI-OBs is reduced compared to that in HC-OBs. Also, the unstable collagen structure is thought to be responsible for the abnormal mineralization, collagen production, and OB differentiation in OI-iPSCs [32,42]. Using OI-iPSCs, we confirmed that genes related to OB differentiation were weakly expressed during the process of in vitro osteogenesis. It is thought to have reduced the number and activity of bone-forming OBs, mechanical strength, and bone mass by producing defective collagen in OB formation. Therefore, these results indicate that in vitro osteogenesis using OI patient-derived iPSCs closely reflects the in vivo characteristics of OI.

In this study, we were able to obtain blood cells from one type I OI patient and confirmed that a T nucleotide was deleted (c.2523delT) from exon 36 of the *COL1A1* gene, replacing the 842nd amino acid (glycine) with alanine and causing a frameshift mutation affecting 266 downstream amino acids. While this mutation is already reported in previous studies [43], the OI patient participating in this study exhibited OI-specific symptoms, such as bone fragility, frequent fractures, and bone deformations, as well as clinical symptoms, such as blue sclera. The expression of type I collagen was altered in the OI patient-derived cells; however, the results differed by the antibodies that were used. While almost no expression of type I collagen was detected using western blot in our first trial (data not shown), we confirmed positive expression in OI in later experiments with a different antibody against type I collagen. The mRNA levels between HC, OI, and editOI showed no significant difference (Figure 4F). Further experimenting is required, but several predictions can be made through these results. Firstly, it can be predicted that the glycine residue replaced by alanine results in a premature termination codon, and the causal variant leads to a frameshift. This may eventually induce the degradation of the mutant mRNA and produce no mutant type I collagen as a result. On the other hand, based on the different outcome, it can be predicted that the mutation results in a simple loss of one allele, which results in partially normal production of type I collagen from the remaining normal allele of *COL1A1* gene. Kawai et al. previously reported osteogenic differentiation and rescue of diseased phenotypes using OI-iPSCs [44]. The authors reported that OI-iPSC-derived osteogenic cells did not show any significant difference of pluripotent markers or osteogenic markers; the amount of calcium deposition was lower in OI-iPSC-derived osteogenic cells. The distribution and amount of type I collagen was also greatly uneven and lower in OI cells. However, the intracellular type I collagen was higher in OI-iPSC-derived osteogenic cells. The authors discussed that, despite sharing a same responsible mutant gene, the heterogeneity may appear in various mutations, and the genotype-phenotype correlation is not yet clear. Our results also may have limitations in showing the exact genotype-phenotype correlation; therefore, further analysis may be critical to fully understand OI, and OI patient-derived iPSCs are still thought as a potential cell source to accomplish this issue.

The attempt to correct the diseased phenotype in OI cells has been previously attempted. Deyle et al. approached this endeavor by inactivating the mutant collagen gene in mesenchymal cells from OI patients and generated iPSCs [28]. MSCs generated from the gene-corrected iPSCs successfully differentiated into osteoblasts. Kawai et al. attempted substituting the mutated nucleotides in OI-iPSCs with wild-type nucleotides by genome editing [43]. The abnormal features that showed during osteogenic differentiation in OI-iPSCs was successfully recovered. Here, we proposed a possible strategy for treating OI using gene-corrected, autologous iPSCs with restored collagen structure. The CRISPR/Cas9 system was used to edit the mutation in the *COL1A1* gene, the causative gene in OI, with the goal of producing iPSCs with normal genes. After gene editing, the same versatility as the parent iPSC control was confirmed in geOI-iPSCs (Figure 3). After selecting the Proto-Spacer Adjacent Motif (PAM) 5′-NGG-3′ sequence recognized by the CRISPR/Cas9 system [45,46], a 20-bp RGEN was constructed starting from the PAM [11]. RGEN 1 and 2 were selected based on off-target removed list 1-0-0, GC content 30–70%, and high out-of-frame scores, indicating that a frameshift is more likely to occur when the indel occurs [14]. RGEN 3 and 4 had higher out-of-frame scores than RGEN 1 and 2. However, T7E1 analysis showed that the PCR bands of RGEN 1 and 2 were clearer than those of RGEN 3 and 4. Therefore, RGEN 1 and 2 were selected for subsequent experiments. An ssODN consisting of a total of (<120 nt) residues was designed and commercially procured (http://www.macrogen.com/ accessed on 3 July 2016) [16]. The prepared donor DNA consisted of a nucleotide sequence spanning a total of 80 residues on both sides of the mutation and the residues predicted to be cleaved by the RGEN [47,48]. As a result, geOI-iPSCs generated OBs with similar morphology and mineralization levels to those observed in OBs differentiated from HC-iPSCs. In addition, the production levels of type I collagen and the restoration of the defected structure were confirmed in geOI-iPSCs (Figure 4). In this study, we differentiated OBs from iPSCs derived from OI patient samples with genetic defects. In conclusion, iPSCs derived from the OI patient sample successfully reflected the in vivo characteristics of OI, such as the abnormal *COL1A1* gene that leads to decreased expression of type I collagen protein in the differentiated OBs. The decreased type I collagen expression in patient cells was recovered by correcting the mutant *COL1A1* gene using the CRISPR/Cas9 system.

## 5. Conclusions

Induced pluripotent stem cells (iPSCs) were generated from an osteogenesis imperfect (OI) patient sample harboring a mutation in the *COL1A1* gene. Using this patient-derived iPSC, the decrease in type I collagen production due to the mutation in the *COL1A1* gene was confirmed along with the reduced mineralization and calcium deposition during in vitro osteogenesis. We successfully restored the expression of type I collagen and the osteogenic ability of the cells by correcting the *COL1A1* gene using the CRISPR/Cas9 system. Taken all together, this study suggests a new possibility of treatment and in vitro disease modeling using patient-derived iPSCs and gene editing with CRISPR/Cas9.

## Figures and Tables

**Figure 1 jcm-10-03141-f001:**
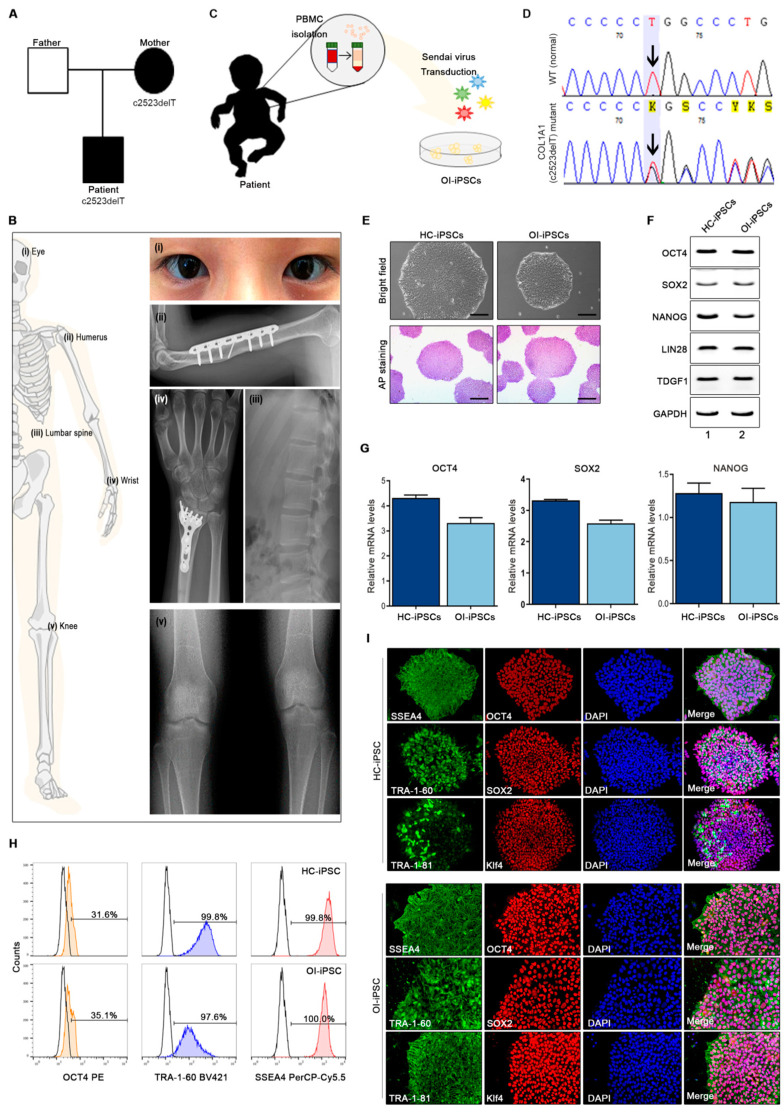
Characterization of iPSCs from healthy control and OI patients. (**A**) Pedigree of the patient and gene sequencing results. (**B**) Clinical phenotypes of OI patients. Transparent blue sclerae caused by abnormal collagen matrix synthesis. Repeated fracture of the humeral shaft. Osteopenic change of carpal bones and repeated fracture of the distal radius. Cortical bone thinning and elevated trabecular bone transparency in the lumbar spines, representing osteopenia. Bone-within-a-bone pattern of vertebral bodies. Thinned cortical bone and reduced bone density. Transverse sclerotic bands in distal femoral and proximal tibial metaphyses. (**C**) PBMCs were isolated from OI patient blood, and iPSCs were generated using Sendai virus. (**D**) Representative DNA sequencing results from OI-iPSCs. (**E**) Morphology of OI and HC-iPSC (upper panel) and alkaline phosphatase (AP) staining (lower panel). (**F**,**G**) Expression of pluripotency markers in HC-iPSCs (*n* = 3) and OI-iPSCs (*n* = 3), as determined by RT-PCR and quantitative RT-PCR. *GAPDH* is a reference gene and was used for quantitative analysis. (**H**) Flow cytometry analysis of cells expressing OCT4, TRA-1-60, and SSEA4. The percentage of OCT4+, TRA-1-60+, and SSEA+ cells in OI and HC-iPSCs. (**I**) Staining for pluripotency transcription factors *Oct3/4, Sox2* and *KLF4* and cell-surface markers *TRA-1-60, TRA-1-81,* and *SSEA4*.

**Figure 2 jcm-10-03141-f002:**
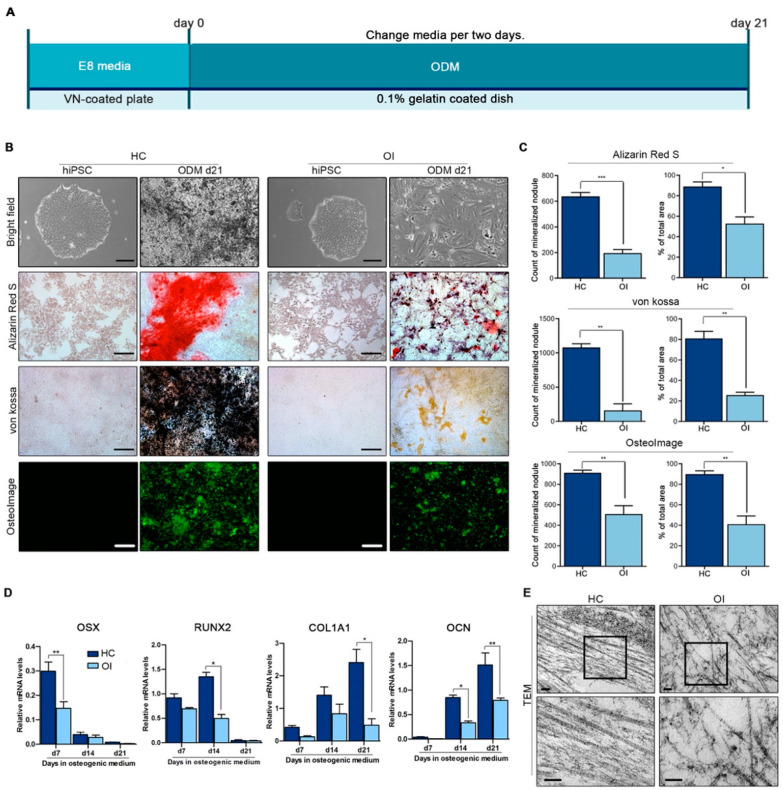
Reduced formation and mineralization of nodules in OI-iPSCs. Differentiation of osteoblasts from OI- and HC-iPSCs. (**A**) Schematic representation of the osteoblast differentiation protocol for OI-, HC-iPS cells by osteogenic differentiation medium (ODM). (**B**) OI-iPSCs and HC-iPSCs were cultured under osteogenic differentiation media and stained after days 7, 14, and 21 with Alizarin Red S and von Kossa to visualize calcium mineral deposition. The amount of hydroxyapatite was measured using the OsteoImage mineralization assay, in which a fluorescent green dye binds to the hydroxyapatite portion of the mineralized matrix (scale bars: 100 µm). (**C**) Quantitative measurements of Alizarin Red S and von Kossa using ImageJ software. The amount of hydroxyapatite was measured using the OsteoImage mineralization assay and quantified using the ImageJ software. Data are means ± SEM. *** *p* < 0.001 vs. OI, ** *p* < 0.005 vs. OI, * *p* < 0.01 vs. OI, indicates statistical significance (Unpaired t-test). (**D**) Quantitation of expression of osteogenic marker genes (OSX, RUNX2, COL1A1, and OCN) in OI-iPSCs after 7, 14, and 21 days of osteogenic differentiation. The graphs show the mean values ± standard deviation of three independent experiments. Data were statistically analyzed using a one-way ANOVA. * *p* < 0.05; ** *p* < 0.01. (**E**) Decrease in collagen production due to abnormal changes in the structure of type I collagen, as confirmed by TEM.

**Figure 3 jcm-10-03141-f003:**
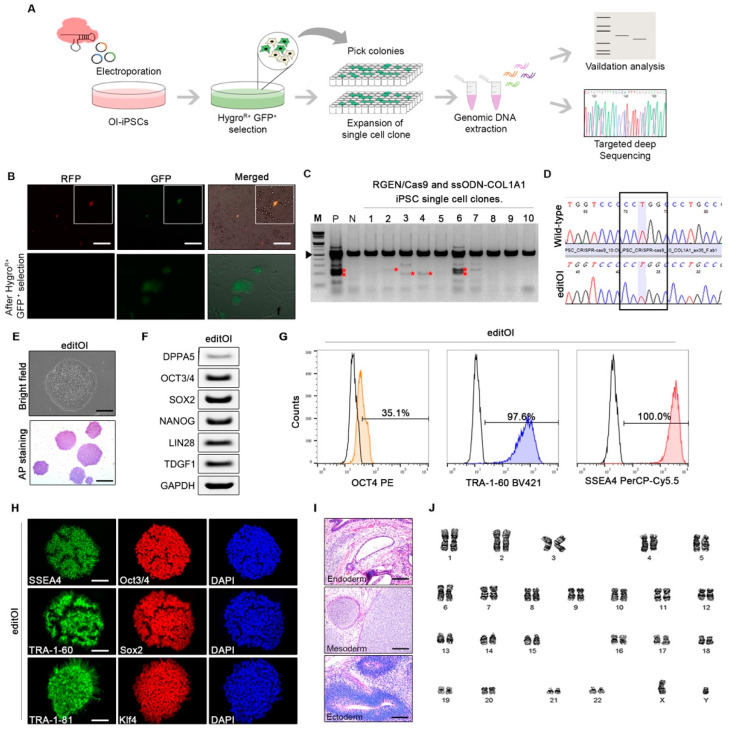
CRISPR/Cas9-mediated editing of the *COL1A1* gene in OI-iPSCs. (**A**) Schematic representation for CRISPR/Cas9-mediated editing of the *COL1A1* gene in OI-iPSCs. (**B**) Fluorescence image of transfected colony. Stable GFP-positive cells were obtained by selection with hygromycin B (scale bars: 100 µm). (**C**) T7E1 assay for insertions and deletions. (**D**) Direct sequencing analysis. We generated a *COL1A1*-corrected iPSC clones of OI patient (editOI-iPSCs). (**E**) Microscopic morphology and AP staining of the *COL1A1*-corrected iPSC clones (editOI-iPSCs) (upper panel; scale bars: 100 µm, lower panel; scale bars: 200 µm). (**F**) Pluripotent marker PCR results and (**G**) flow cytometry analysis of cells expressing *OCT4, TRA-1-60*, and *SSEA4.* (**H**) Immunofluorescence image of editOI-iPSCs analyzed for pluripotency markers (scale bars: 100 µm). (**I**) editOI-iPSC teratoma histology (scale bars: 100 µm). (**J**) Morphology and karyotype of editOI-iPSCs.

**Figure 4 jcm-10-03141-f004:**
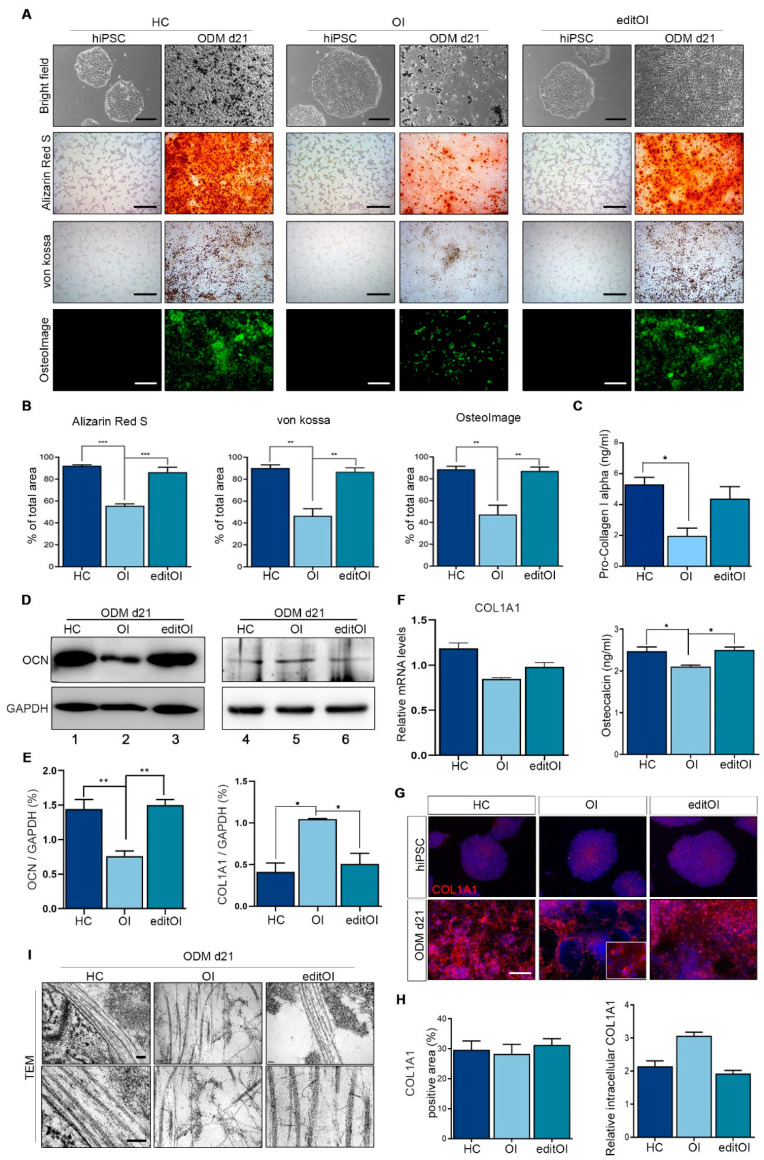
Improvement of collagen structure stability and osteoblast differentiation of edited OI-iPSCs. editOI-iPSCs, OI-iPSCs, and HC-iPSCs were cultured in osteogenic differentiation medium for 21 days. (**A**) Observation of morphological features by bright-field microscopy. Osteoblastic (calcium phosphate) deposition by osteoblasts was measured by Alizarin Red S and von Kossa staining. The amount of hydroxyapatite was measured using the OsteoImage mineralization assay (scale bars: 100 µm). (**B**) Quantitative measurements of Alizarin Red S and von kossa using ImageJ software. The amount of hydroxyapatite was measured using the OsteoImage mineralization assay and quantified using the ImageJ software. The graphs show the mean values ± standard deviation of three independent experiments. Data were statistically analyzed using a one-way ANOVA. * *p* < 0.05; ** *p* < 0.01; *** *p* < 0.001. (**C**) Pro-Collagen I alpha and Osteocalcin levels were measured in the osteogenic differentiation supernatant. The graphs show the mean values ± standard deviation of three independent experiments. Data were statistically analyzed using a one-way ANOVA. * *p* < 0.05. (**D**) Representative western blot analysis of COL1A1 and OCN in cell lysates, with quantification. GAPDH was used as a loading control. (**E**) Western blot band quantification through image analysis software. Data were statistically analyzed using a one-way ANOVA. * *p* < 0.05; ** *p* < 0.01. (**F**) Quantitation of expression of osteogenic marker genes (COL1A1) in HC, OI, and editOI-iPSCs after 21 days of osteogenic differentiation. Data were statistically analyzed using a one-way ANOVA. * *p* < 0.05. (**G**) Representative images of immunostaining of type I collagen in HC-iPSCs, OI-iPSCs, editOI-iPSCs (scale bars: 100 µm). (**H**) Percentage of type I collagen positive area, as assessed using the ImageJ software. Quantification of intracellular type I collagen. The graphs show the mean values ± standard deviation of three independent experiments. Data were statistically analyzed using a one-way ANOVA. * *p* < 0.05. (**I**) Decrease in collagen production due to abnormal changes in the structure of type I collagen was confirmed by TEM images (scale bars: 300 nm).

**Figure 5 jcm-10-03141-f005:**
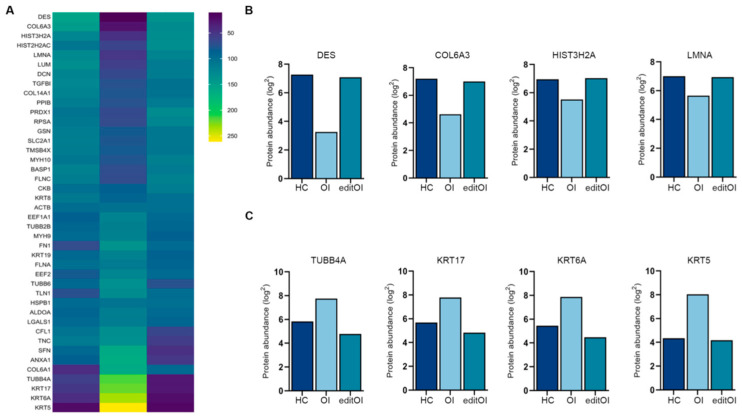
Label-free quantitative analysis of OI-OBs and editOI-OBs. (**A**) Hierarchical clustering of the differentially expressed proteins. Yellow denotes high relative expression, and blue denotes low relative expression. (**B**,**C**) Four proteins with high Score Sequest HT in HC and editOI-OBs and four proteins with high Score Sequest HT in OI-OBs were identified.

**Table 1 jcm-10-03141-t001:** Off-target information.

RGEN	Target Sequence	GC Contents (%)	Out-of-Frame Score	0 bp Mismatch	1 bp Mismatch	2 bp Mismatch
h*COL1A1*_RG1	AAAGGCGAACCTGGTGATGCTGG	21	55	1	0	0
h*COL1A1*_RG2	GCTGGTGCTAAAGGCGATGCTGG	30.3	60	1	0	0
h*COL1A1*_RG3	GGGGTCCAGCGGGTCCGGCAGGG	41	80	1	0	0
h*COL1A1*_RG4	GGGGGTCCAGCGGGTCCGGCAGG	41.5	85	1	0	0

**Table 2 jcm-10-03141-t002:** List of PCR primer sequences used for RT-*PCR* and qRT-PCR analysis.

Target Name	Direction	Primer Sequence	Target Size (bp)	Gene Reference
human*OCT3/4*	Forward	ACCCCTGGTGCCGTGAA	190	NM_001173531.1
	Reverse	GGCTGAATACCTTCCCAAATA		
human*SOX2*	Forward	CAGCGCATGGACAGTTAC	321	NM_003106.3
	Reverse	GGAGTGGGAGGAAGAGGT		
human*LIN28*	Forward	GTTCGGCTTCCTGTCCAT	122	NM_024674.4
	Reverse	CTGCCTCACCCTCCTTCA		
human*NANOG*	Forward	AAAGGCAAACAACCCACT	270	NM_024865.2
	Reverse	GCTATTCTTCGGCCAGTT		
human*TDGF1*	Forward	TCCTTCTACGGACGGAACTG	140	NM_003212.3
	Reverse	AGAAATGCCTGAGGAAAGCA		
humanDPPA5	Forward	CGGCTGCTGAAAGCCATTTT	215	NM_001025290.2
	Reverse	AGTTTGAGCATCCCTCGCTC		
human*RUNX2*	Forward	AGTGGACCCTTCCAGACCAG	261	NM_001015051.3
	Reverse	ATGGTCGCCAAACAGATTCA		
humanCOL1A1	Forward	CAGGGTGTTCCTGGAGACCT	291	NM_000088.3
	Reverse	AGGAGAGCCATCAGCACCTT		
human*OCN*	Forward	CCAGGCGCTACCTGTATCAA	231	NM_199173.5
	Reverse	AGGGGAAGAGGAAAGAAGGG		
human*GAPDH*	Forward	GAATGGGCAGCCGTTAGGAA	414	NM_002046
	Reverse	GACTCCACGACGTACTCAGC		

**Table 3 jcm-10-03141-t003:** Osteogenesis imperfecta cell lines with mutations in COL1A1.

OI type	Type	Exon	Nucleotide Change	Protein Mutation
I	DNA mutation	36	c.2523delT	p.Gly842Alafs *

* Alanine Frameshift.

**Table 4 jcm-10-03141-t004:** sgRNA target site sequences.

Target Name	Target Sequence	Target Size (bp)	Tm
*COL1A1*_ex36_F	CCCCCATCATTTTTCATCAC	477	60
*COL1A1*_ex36_R	CAGAGAGGCGGGTGATACTC		

## Data Availability

Data is contained within the article.

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
