# Peer review of "Restoration of Osteogenesis by CRISPR/Cas9 Genome Editing of the Mutated COL1A1 Gene in Osteogenesis Imperfecta"

_jcm, 2021, doi:10.3390/jcm10143141_

Round 1

Reviewer 1 Report

Although the revised manuscript improved in several points, authors ignored two important comments of this reviewer.

  1. I requested to perform mRNA analyses of COL1A1 gene to investigate whether the frameshift mutation produced truncated transcript or no product due to nonsense-mediated mRNA decay. This is a critical issue in OI cases with frameshift mutations, and it is important to make it clear whether this mutation results in simple loss of one allele or has a dominant negative effect like those of glycine substituted mutations. Although authors did not describe the clinical classification of the patient, this patient may belong to type I based on photos and figures, suggesting that the frameshift mutation in this case resulted in simple loss of one allele. If so, the complete loss of COL1 in Figure 4F can not be explained, because the normal COL1A1 will be produced from the remaining normal allele of COL1A1 gene. Authors should analyze the mRNA and discuss with this point.
  2. Author ignored to refer following previous study.
    Kawai et al. Nature Biomed Eng 2019;3:558-70.

Reviewer 2 Report

This reviewer agrees that the authors present their modelling data as a supplemental figure, and as stated, only as a peripheral methodology for analysis not directly related to their own results. The authors must make sure that the manuscript with final figures are adjusted accordingly, as the current form of the manuscript still seemed to contain the modelling within the figure 4, and not incorporated in the new supplemental data.

Regarding the new COL1A1 analysis, it is fine, but the statistics are missing and must be presented.

Round 2

Reviewer 1 Report

no further comments

This manuscript is a resubmission of an earlier submission. The following is a list of the peer review reports and author responses from that submission.

Round 1

Reviewer 1 Report

In this paper, Jung et al. demonstrated the in vitro disease modeling of OI by patient-derived iPS cells. They generated iPS cells from an OI patient with a frame-shift mutation of the COL1A1gene, and osteogenic differentiation of this iPS cell was induced along with wild-type iPS cell. The results indicated that OI-derived iPS cells showed lower differentiation property by the expression of differentiation-related gene and the formation of mineralized nodules.  Using in silico analyses, they predicted abnormal collagen structure and demonstrated it by TEM.  Then using CRISPR-cas9 method, they established gene-editing iPS cells, and showed that the gene-editing restored the osteogenic differentiation property and also identified several protein of which the expression was correlated with the COL1A1 mutation.

This case report showed the utility of disease-specific iPS cells for the analyses of hereditary rare disease, but the almost all contents of in this study was previously reported (authors failed to refer a previous report Nat Biomed Eng. 2019;3:558-70). Although many experiments were done in this study, it has little scientific significance.

Comments

  1. How did authors identify the mutation in this patient? Gly842 is in exon 37 not 36. Mutation should be described Gly842Alafs*266. This is a flame-shift mutation which causes a premature stop codon in the downstream residue. Therefore it is essential to demonstrate the expression of mRNA from each allele, and the presence or absence of NMD.
  2. The differentiation method of OB from iPS cells used in this study was not previously reported and therefore should be described more precisely. It looks a very simple method but is it a reproducible method?
  3. In any RT-PCR analyses, there were no description for the meaning of “relative mRNA level”. Without a proper control, it is difficult to evaluate the data.
  4. Immunostaining of COL1A is not clear and should be replaced with better one.
  5. It was a surprise that there were no expression COL1 in OI sample by the western blot. How do authors explain this result?
  6. In silico structure analyses are interesting approaches. However, in line 95 author described that mutant structure was generated by mutating residue 842, glycine 842 to alanin 842 of the wild-type structure. If this is correct, this analysis has no meaning for the data of this case, because flame-shift mutation changed amino acids not only at G842 but also following 265 amino acids.
  7. There is poor description of proteome analysis. What is the sample of each cells, iPS cells or OB derived from iPS cells? What is the purpose of this experiment? Why proteome and not transcriptome? Why there are no discussion about the results of this experiment? .

Reviewer 2 Report

jcm-899934

The manuscript by Jung et al. presents an interesting gene-editing approach to correct the mutation in the COL1A1 gene in induced pluripotent stem cells from an individual with OI. The authors obtained peripheral blood monocytes from one OI patient carrying a single deletion in COL1A1. They used a standard viral based delivery method to derived pluripotent stem cells, and study their differentiation behavior under osteogenic conditions. They went further and used CRISPR-Cas9 to correct the genetic defect in the iPSCs and showed that osteogenesis was restored, based on limited outcomes. The manuscript is relatively well written, but there are some typos and discrepancies between the text and the figures. In addition, there are many issues (listed below) that must be addressed.

Line 87: define E8 medium

Line 108: RGEN should be defined here. Delete ‘used’ in the following “…were designed using the used ToolGen optimized…’

Line 110: the T7E1 assay should be briefly described.

Line 116: delete sgRNA

Line 125: Alizarin red S staining measures calcium deposition, and only indirectly assesses OB terminal differentiation.

Line 145. The table listing the primers used is not Table 2 but Table 4. Must be adjusted throughout.

Line 186: number of cells should read 1 X 106

Line 201: ‘Error bars represent the standard error of the mean.’ Can be deleted (redundant).

Line 209: It should be mentioned that this mutation was previously reported in other patients.

Line 213: correct the following: ‘deformations which WERE observed through X-ray’

Lines 217-218: Discrepancy between the text narrative and the Figure 1 legend. Please validate the origin of the material used for the DNA sequencing (patient tissue/cells, or OI-iPSCs).

Lines 222-223: The RT-qPCR data for OCT4 and SOX2 seem to be slightly decreased in the OI-iPSCs. Please explain if statistical analysis was done or not. Also, was this analysis performed on a pool of colonies, or from several individual colonies. Depending on the answer, the statistical test could be applicable (n>3) or not (n=1).

Line 226: Klf4 should be capitalized (KLF4) according to nomenclature. Also, throughout the text, gene names and mRNA should be appear italicized/capitalized (COL1A1).

Line 228: Expression levels by IF is not quantitative. It should be mentioned instead that the 'intensity levels for the markers did not differ markedly'. Also, there is typo for ‘Comclusion’.

Line 241: The X-axis labeling for the RT-qPCR control iPSCs are wrongly labeled as 'HCliPSCs'.

Line 244: '...the collagen structure produced in the OI patient.'

Line 245: The modeling is interesting, but it should be clearly mentioned whether the mutation causes 'haploinsufficiency' or if the mutant collagen carrying the mutation is produced and incorporated within the triple helix. It would be presumed that since the c-propeptide would not be made, it is unlikely the mutant collagen molecule can incorporate. If true, the modeling is useless and must be omitted from the manuscript. This information is also very pertinent to the interpretation of the data presented after and in the discussion.

Line 271: The statistical analysis shown for COL1A1 expression and comparing day 7 to day 14 (as NS) would not be possible with a T-test. Only ANOVA could yield such a comparison. Adjust or remove. This applies also for the quantification of the mineralization as shown in the supplemental data. Also, it would be important to mention that the RT-qPCR assay detects both the WT and mutant COL1A1 mRNA, and does not discriminate between the 2 forms.

Lines 275-290: The various panels within legend of Figure 2 are mislabeled - re-ordering needed to harmonize with the corresponding text section.

Line 294: The table 1 could be deleted. The description of the patient genotype in Figure 1 and corresponding text is sufficient.

Line 299: The rationale for using the RFP and GFP markers should be better described within the legend. The data in panel D showing RFP and GFP fluorescence is not convincing and could perhaps be improved by enhancing contrast or showing higher magnification of positive cells.

Line 306: In the legend to figure S3, it is indicated that the images are from 293T cells, not iPSCs.

Line 319: In the methods, the description of the in vivo procedure in the mouse is different (subcutaneous versus intraperitoneal)?

Lines 335-336: Assuming the scale bars are the same for the bright field and AP staining, why are the colonies smaller for AP? Are these representative images?

Lines 341-380: the entire paragraph is referring to Supplementary Figure S3 while it should be S4.

Lines 362-363: It is unclear why a marker of bone 'resorption' (DPD) would have been included for the analysis. The OBs cultures would certainly not contain 'resorbing' activity. Gene expression markers in the differentiated day 21 OBs, like those presented under figure 1, would have been good instead.

Lines 359-360: It would be odd to have so little reduction of the measured OCN levels in the media of OI OBs, while there was such a dramatic reduction in the mRNA expression, as shown in figure 2G. Authors should provide explanation.

Lines 368-369: It is striking that there is an almost total absence of COL1A1 in the OI-iPSCs. It would have been good to have the protein levels in the iPSCs differentiated in osteoblasts instead. Is it possible that the anti-COL1A1 antibody recognizes only the c-terminal portion of the WT COL1A1, which would be absent in the OI?

Line 398: The methods describing the proteomic analysis are missing from the manuscript. Possible implications of dataset are missing.

Line 402: The Appendix table for the proteomics is missing.

Discussion: authors should take into considerations the above-mentioned comments, more particularly the mutant collagen production (or not), and put that into the context of their data and interpretation. Further, it would be interesting if they could compare the technology they used in light of other studies in the field, for example using RNA interference, or other methodologies to knockdown mutant COL1A1 in OI. Also, what could be the future implications and translational aspects of their work.

Reviewer 3 Report

The manuscript is well-written and presents an important study for OI treatment. The study is well-conducted and figures are informative and of high quality.

A minor comment is a concern regarding the structure of collagen. The authors investigate the structure of collagen with regard to the causal variant. While perfectly true that it is predicted that the glycine residu is replaced by alanine and resulting in a premature termination codon, the causal variant leads to a frameshift and degradation of the mutant mRNA. As such, no mutant collagen will be produced because the mutant mRNA will be degraded. The abnormality in the structure of collagen that is observed in the TEM studies for example is therefore more related to the decrease in collagen production rather than related to the production of mutant collagen (also if mutant collagen would be produced this would lead to a more severe clinical phenotype). As such, the authors might alter the different paragraphs where the structure of collagen is discussed.

Round 2

Reviewer 2 Report

The authors addressed most concerns raised adequately. However, there are 2 issues raised by the Reviewer that were not considered appropriately (see below prior comment, answer, and reply by Reviewer):

Line 245: The modeling is interesting, but it should be clearly mentioned whether the mutation causes 'haploinsufficiency' or if the mutant collagen carrying the mutation is produced and incorporated within the triple helix. It would be presumed that since the c-propeptide would not be made, it is unlikely the mutant collagen molecule can incorporate. If true, the modeling is useless and must be omitted from the manuscript. This information is also very pertinent to the interpretation of the data presented after and in the discussion.

Answer) Osteogenesis imperfecta (OI) type I is usually caused by COL1A1 stop or frameshift mutations, leading to COL1A1 haploinsufficiency. These mutations usually create a premature termination codon. With normal alpha chains of type I collagen being produced from the wild-type allele, haploinsufficiency generally leads to a mild OI phenotype [6]. Accordingly, we changed the misleading results in the manuscript. “As a result, a heterozygous frame shift mutation has been identified in COL1A1, which may lead to haploinsufficiency.”

Reply by Reviewer: Because the mutation in COL1A1 causes haploinsufficiency, and that the mutant allele does not contribute to production of altered COL1A1 protein, the modeling, as stated previously, is totally hypothetical and must be removed from the manuscript. The gene editing correction of the mutation simply restore normal abundance of triple helical COL1 protein, it does not normalize its structure.

Lines 362-363: It is unclear why a marker of bone 'resorption' (DPD) would have been included for the analysis. The OBs cultures would certainly not contain 'resorbing' activity. Gene expression markers in the differentiated day 21 OBs, like those presented under figure 1, would have been good instead.

Answer) Thanks for your kind comment. We measured representative OCN and DPD as biochemical markers of bone metabolism, to observe the changes in biochemical bone metabolism caused by collagen mutations. Measurements of DPD levels identify mineral homeostasis defects, and through this results we observed increased levels of DPD in gene-edited OI compared to OI, and previous studies done by R Morello et al. showed that increasing DPD levels may reduce bone synthesis [7]. We have added this description in the manuscript and cited this reference.

Reply by Reviewer: The authors have misinterpreted the data contained in the reference provided about deoxypyridinoline crosslinks (DPD) levels. DPD is a marker of bone resorption upon the degradation of COL1 by osteoclasts, and normally measured in the urine of patients/animals. That is what was done in the Morello et al. paper, looking at DPD in the urine samples from mutant mice. This was done to conclude that resorption activity by osteoclasts was not altered in the Crtap KO mice, certainly not to infer increased DPD levels would impact bone synthesis. The authors need to remove the DPD data from the manuscript and edit the text accordingly.
